# Conditional Generative Quantile Networks via Optimal Transport and Convex Potentials

## Abstract

Quantile regression has a natural extension to generative modelling by leveraging a stronger convergence in point rather than in distribution. While the pinball quantile loss works in the scalar case, it does not have a provable extension to the vector case. In this work, we consider a quantile approach to generative modelling using optimal transport with provable guarantees. We suggest and prove that by optimizing smooth functions with respect to the dual of the correlation maximization problem, the optimum is convex almost surely and hence construct a Brenier map as our generative quantile network. Furthermore, we introduce conditional generative modelling with a Kantorovich dual objective by constructing an affine latent model with respect to the covariates. Through extensive experiments on synthetic and real datasets for conditional generative and probabilistic forecasting tasks, we demonstrate the efficacy and versatility of our theoretically motivated model as a distribution estimator and conditioner.

## 1 Introduction

A rudimentary result in probability theory states that applying the cumulative distribution function (CDF) to the underlying continuous random variable results in a uniformly random sample from the unit interval. Correspondingly, passing a uniformly random sample from the unit interval through the inverse CDF, a.k.a. the univariate quantile function, results in a random sample from the desired CDF, a simple fact that has been exploited in simulating many distributions (Devroye, 1986). Quantile functions also play a pivotal role in measuring financial risks (Rüschendorf, 2013) and in building regression functions at various levels (Koenker & Gilbert, 1978) and robust statistical estimators under model mis-specification (Koenker, 2005). The pin-ball loss in quantile regression has also been used in training supervised classifiers (Huang et al., 2014) and more recently in building unsupervised generative models (Ostrovski et al., 2018).

The perspective to treat quantile functions as a natural mathematical object for modern generative models is very appealing (Parzen, 1979). Traditional quantile regressors are trained at fixed quantile levels, and the entire quantile function may be obtained through interpolation or smoothing, which, however, suffers from the so-called quantile crossing problem, namely that the monotonicity property of the quantile function is often violated due to training quantile regressors at different levels independently. Despite this nuance, the quantile approach, in its the univariate form, has been successfully applied in many applications (Koenker, 2005) and more recently in probabilistic forecasting (e.g. Gasthaus et al., 2019; Gouttes et al., 2021). However, the pinball loss, while effective for univariate datasets, does not extend immediately to multivariate responses, a setting of paramount practical importance and is our focus in this work. While in principle it is possible to apply quantile regression to each dimension, as in Ostrovski et al. (2018), this naive approach ignores the high-order correlations between components and can lead to incorrectly learned models.

Recently, optimal transport has garnered substantial interest in generative modelling (Arjovsky et al., 2017; Makkuva et al., 2020; Finlay et al., 2020; Huang et al., 2021), domain adaptation (Courty et al., 2017), imaging (Bonneel et al., 2015), semi-supervised learning (Solomon et al., 2014), and much more (Peyré & Cuturi, 2019). The main goal of optimal transport is to learn a coupling between two distributions such that we can transport mass from one distribution to another at minimal cost. In fact, under mild assumptions, the optimal coupling between two continuous distributions can be obtained through the gradient of a convex function (Brenier, 1991; McCann, 1995). Since gradients

of convex function are monotonic, and hence coincide with quantile function in the univariate setting. This fact motivates attempts, such as here, to construct multivariate quantile functions through the gradient of convex functions. Compared to other approaches for generative modelling, quantile networks have the added advantage of exact estimation of quantile levels. Indeed, a pointwise estimation of quantile levels removes the necessity for empirical quantile esimation of non-quantile based generative models through say MCMC (Rasul et al., 2021).

In this work, we seek to generalize generative quantile modelling to the multivariate setting and we extend the vector quantile regression of Carlier et al. (2016) to conditional generative quantile modelling. Our main contributions are summarized as follows,

- We construct a dual formulation of correlation maximization and extend its usage to generative quantile networks that is highly theoretically motivated.

- To the best of our knowledge, we are the first work to propose conditional generative modelling using a Kantorovich dual objective.

- Previous works that are based on the Kantorovich dual restrict their models to input convex neural networks (Amos et al., 2017, ICNNs). Instead, we relax this assumption by parameterizing our model as smooth neural networks and verify the sufficiency and advantage of this relaxation.

- We further argue that convex potential gradients are natural quantile networks since they automatically satisfy the non-crossing property of quantile functions.

In addition to the aforementioned novelties, our proposed conditional generative quantile network is also very flexible and versatile. In particular, we conduct thorough experiments to show that it can serve as a conditioner for generative tasks and can also be used for probabilistic forecasting.

## 2   BACKGROUND

In this section, we recall some background on generative quantile functions and optimal transport. In the univariate case, given a cumulative distribution function (CDF) $F(x) = \Pr(X \leq x)$ of random variable $X$, the corresponding quantile function $Q$ is defined as: $\forall u \in (0, 1)$,

$$Q(u) = F^{-1}(u) := \inf\{x : F(x) \geq u\}. \tag{1}$$

While the CDF can be straightforwardly extended to the multivariate case, the quantile function, its inverse, is less obvious as the loss of the total ordering on the real line made it impossible to "just invert" $F$. Instead, we can think of the quantile function $Q$ as a mapping that transforms a uniformly random variable $U$ to a sample from the CDF $F$ of interest, be it univariate or not. This generative view of the quantile function turns out to be quite fruitful and allows us to learn distributions via quantile regression, through for instance the pinball loss $\ell_u$ at a fixed quantile level $u \in (0, 1)$:

$$\ell_u(q; Y) = u(Y - q)^+ + (u - 1)(Y - q)^-. \tag{2}$$

Indeed, for fixed $u$, minimizing the expected loss $\mathbb{E}[\ell_u(q; Y)]$ yields exactly $q = Q(u)$ for the quantifule function $Q$ of $Y$ (Koenker & Gilbert, 1978).

**Generative Quantile Networks.** Now, to estimate the entire quantile function $Q$, we can simply replace the fixed quantile level $u$ with a univariate random variable $U \sim \mathrm{Unif}([0, 1])$. We can now minimize the expected loss $\mathbb{E}\ell_U(Q(U); Y)$ w.r.t. $Q$ to learn the entire quantile function. The univariate quantile loss has been applied successfully recently in probabilistic time-series forecasting. For instance, Gouttes et al. (2021) proposed stacking an implicit quantile network on top of an recurrent neural network (RNN) to model the temporal dynamics of the covariates. By minimizing the quantile loss, exact point-wise quantiles can be inferred when forecasting.

However, when we transition to the multivariate setting where $U$ is uniform over the hypercube $[0, 1]^d$, we can no longer directly minimize the expected loss $\mathbb{E}\ell_U$. A natural alternative is to minimize the $L_p$ norm of the vector quantity in eq. (2). For instance, the autoregressive implicit quantile networks (AIQN) of Ostrovski et al. (2018) amounts to adopting the $L_1$ norm. Wen & Torkkola (2019) proposed to train a Gaussian-copula model jointly on top of a generative multivariate quantile model with the $L_1$-norm quantile loss and account for higher-order correlations in a two-step procedure. Normalizing flows are a popular class of generative models that allow explicit evaluation

of the likelihood and the inverse of Jacobian. In particular, Wang et al. (2019) parameterized their quantile function with an increasing triangular map and applied it to novelty detection. However, an extension to conditional quantile estimation was not considered.

**Optimal Transport.** Consider two probability measures $\mu$ and $\nu$ on $\mathcal{X} \subseteq \mathbb{R}^d$ and $\mathcal{Y} \subseteq \mathbb{R}^m$, respectively, and $c : \mathcal{X} \times \mathcal{Y} \to [0, +\infty]$ a (closed) cost function. The Monge problem refers to finding a transport map $G : \mathcal{X} \to \mathcal{Y}$ that minimizes the total transport cost:

$$C(\mu, \nu) := \inf_{G:\mathcal{X}\to\mathcal{Y}} \left\{ \int_{\mathcal{X}} c(x, G(x)) \mathrm{d}\mu(x) \mid G_{\#}\mu = \nu \right\}, \tag{3}$$

where $G_{\#}\mu$ denotes the push-forward measure by $G$. An ubiquitous transport cost is the squared Euclidean distance $c(x, y) = \|x - y\|_2^2$, yielding the (squared) 2-Wasserstein distance, i.e. $W^2(\mu, \nu)$. A common issue with optimal transport in practice is its computation in high dimensions. Kolouri et al. (2019), among many others, propose to employ random projections and to reduce to the univariate setting with provable convergence guarantees. However, the number of projections sampled from the unit hypersphere necessarily scales with the dimensions of the distribution. Alternatively, Kantorovich's duality offers an objective that can be conveniently approximated by restricting the structure of dual potentials. Formally, the Kantorovich dual problem is given by

$$\sup_{(\varphi,\psi)\in L^1(\mu)\times L^1(\nu)} \int_X \varphi \mathrm{d}\mu(x) + \int_Y \psi \mathrm{d}\nu(y) \text{ s.t. } \forall x \in \mathcal{X}, \forall y \in \mathcal{Y}, \ \varphi(x) + \psi(y) \le c(x, y) \tag{4}$$

where $\varphi \in L^1(\mu), \psi \in L^1(\nu)$ can be chosen to be bounded and continuous (Villani, 2008). In fact, should the dual admit a solution, $\frac{1}{2}\| \cdot \|_2^2 - \varphi$ and $\frac{1}{2}\| \cdot \|_2^2 - \psi$ are closed convex functions and are Fenchel conjugates of each other, provided that $c$ is the squared Euclidean distance. A byproduct of this is that the gradient map $\mathrm{Id} - \partial\varphi$ is an optimal coupling between $\mu$ and $\nu$ (Villani, 2008).

Makkuva et al. (2020) proposed parameterizing $\frac{1}{2}\| \cdot \|_2^2 - \varphi$ as an Input Convex Neural Network (ICNN) (Amos et al., 2017) and optimize the Kantorovich dual for generative modelling tasks. Huang et al. (2021) utilized Brenier's theorem to introduce Convex-Potential Flows (CP-Flows) in which they restrict their flow model with strong convexity. Thus, the flow is the convex gradient that is invertible numerically using a convex solver and they use Hutchinson's trace estimator to estimate the logarithmic Jacobian. Although similar to ours, both works are not conditional and neither are they generative quantile networks since the underlying distribution is not learned through all quantiles jointly. Furthermore, we do not constrain our model to be convex. We show in our work that by optimizing smooth neural networks with our correlation maximization objective below, the dual admits a convex solution.

## 3 CONVEX POTENTIALS AS GENERATIVE QUANTILE NETWORKS

We propose a novel method for density estimation using a quantile approach that generalizes to the multivariate setting and allows for flexible task-dependent conditioning. Briefly, we first formulate the dual problem of correlation maximization as our multivariate quantile loss. Then, we construct an affine latent model with respect to the covariate embeddings that is highly versatile. In particular, any task-dependent deep feature extractor can be used to extract latent embeddings of the covariates and can be jointly trained with our quantile objective without any auxiliary regularization. We then provide theoretical guarantees of convergence to an optimal convex function.

### 3.1 MULTIVARIATE QUANTILES AS CONVEX POTENTIALS

Before we consider the multivariate case, let us first analyze univariate quantile functions to motivate the correlation maximization problem. Let $Q_Y(u) : [0, 1] \to \mathbb{R}$ be defined as an univariate quantile function of $Y \sim \nu := \mathrm{Law}(Y)$. By definition, quantile functions are monotonically increasing so $Q_Y(u)$ will be correlated with $u$. A quantile function is thus a solution to the correlation maximization problem, as established in the following equality (Galichon & Henry, 2012):

$$\int_0^1 u Q_Y(u) \mathrm{d}(u) = \max \left\{ \mathbb{E}(UY), U \sim \mathrm{Unif}([0, 1]) \right\}. \tag{5}$$

Now, we are interested in generalizing this equality to the multivariate setting for $Y \in \mathbb{R}^d$ and $U \sim \text{Unif}([0,1]^d)$. For the remainder of this paper, $\mu$ and $\nu$ are $d$-dimensional distributions. Formally, we start with the definition of maximal correlation functionals that motivates our goal of using convex potential gradients to parameterize multivariate quantile functions. Recall that $L_d^p$ denotes the $L^p$ space with range space $\mathbb{R}^d$.

**Definition 1** (Maximal Correlation Functionals, Galichon & Henry 2012). *A functional $\rho_\mu : L_d^2 \to \mathbb{R}$ is called a maximal correlation functional with respect to distribution $\mu$ if for all $Y \in L_d^2$,*

$$\rho_\mu(Y) := \sup \left\{ \mathbb{E}[U^\top Y], U \sim \mu \right\}. \tag{6}$$

Should the distribution $\mu$ be absolutely continuous w.r.t. the Lebesgue measure, then there exists a closed convex function $f : \mathbb{R}^d \to \mathbb{R}$ such that $Y = \nabla f(U)$ $\mu-$almost surely (Villani, 2008). Indeed, it is clear that under these mild assumptions on $\mu$, we can establish a multivariate extension of (5) using the convex potential gradient of $f$. Furthermore, $\nabla f$ achieves an optimal coupling between $\mu$ and $\nu$ (Huang et al., 2021) such that $\rho_\mu(X) = \left\{ \mathbb{E}[U^\top \nabla f(U)], U \sim \mu \right\}$. By Brenier's theorem, the existence and uniqueness of the convex potential gradient map between $\mu$ and $\nu$ can be established.

**Theorem 1** (Brenier 1991). *If $Y$ is a squared-integrable random vector in $\mathbb{R}^d$, there is a unique map of the form $T = \nabla f$ for some convex function $f$ such that $\nabla f_{\#}\mu = \nu$.*

In other words, we say that $\nabla f$ is a Brenier mapping between $\mu$ and $\nu$. Now, we are ready to define the generalized quantile function as a convex potential gradient.

**Definition 2** (Convex Potential Quantile, Chernozhukov et al. 2017; Hallin et al. 2021). *Let $f : \mathbb{R}^d \to \mathbb{R}$ be a closed convex function. Assume $\mu$ is absolutely continuous on $\mathbb{R}^d$ w.r.t. the Lebesgue measure. Then, the Convex Potential Quantile of $Y \sim \nu$ is defined as $Y = \nabla f(U)$ where $U \sim \mu := \text{Unif}([0,1]^d)$, i.e. $\nabla f_{\#}\mu = \nu$.*

Having adopted the above convex potential quantile function as a natural generalization to the multivariate setting, we now turn to its estimation. It turns out that directly addressing the maximal correlation functional is a bit challenging. Instead, we resort to its dual problem to learn the Brenier mapping $\nabla f$. Revisiting the primal where we have also introduced a covariate $X$:

$$\sup\{\mathbb{E}(U^\top Y), U \sim \mu, \mathbb{E}(X|U) = \mathbb{E}(X) = 0\}. \tag{7}$$

Here the mean-independence constraint is added to decorrelate $U$ with the covariate $X$. The dual is derived as the following:

$$\inf_{(\varphi,\psi,b)} \int_{[0,1]^d} \varphi \mathrm{d}\mu(U) + \int_{\mathbb{R}^d} \psi \mathrm{d}\nu(Y) \ \text{s.t.} \ \varphi(U) + \psi(Y) \geq U^\top Y \tag{8}$$

$$\mathbb{E}(X|U) = \mathbb{E}(X) = 0, \tag{9}$$

where $\varphi$ is smooth. We define $\psi$ as the Legendre transformation of $\varphi$,

$$\psi(y) := \sup_{U \in [0,1]^d} \{U^\top Y - \varphi(U)\}, \tag{10}$$

to reduce the problem down to optimizing only $\varphi$ and $b$ as we expect the optimum $\varphi$ to satisfy $\varphi^* = \psi$. By complementary slackness, the inequality in eq. (8) becomes equality for optimal $U$:

$$\varphi(U) + \psi(Y) = U^\top Y. \tag{11}$$

By differentiating eq. (11) w.r.t $U$, we arrive at $Y = \nabla \varphi(U)$. However, this alone is not sufficient for characterizing the existence and convergence of smooth function $\varphi$ to the optimal convex potential gradient that couples $\mu$ and $\nu$. In section 4, we provide theoretical justification and necessary conditions for existence and convergence.

## 3.2 CONDITIONAL GENERATIVE QUANTILE NETWORKS

Suppose an explanatory vector $X \in \mathbb{R}^m$ is given, and we are interested in generating samples $Y$ given $X$. Indeed, the target distribution is now the joint distribution of $(X, Y)$, $\nu := \text{Law}(X, Y)$.

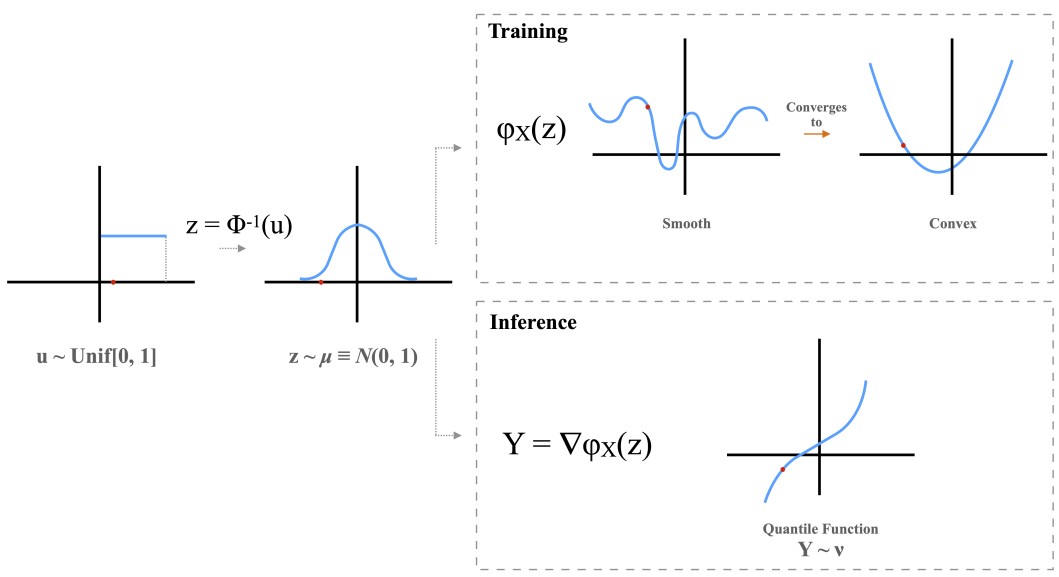

Figure 1: **Training and inference pipeline of our proposed conditional generative quantile network for a univariate component.** The quantile level $u$ is sampled from the unit interval and pushed into standard normal. During training, our smooth parameterization learns an optimal coupling between the standard normal and the target distribution, and converges to a convex potential. During inference, the gradients are computed via auto-differentiation to generate the respective quantile $Y \sim \nu$. The convex potential gradient $\nabla \varphi_X$ is a comonotonic push-forward mapping of $\mu$ to $\nu$.

Carlier et al. (2016) have shown that a convex function can be approximated via an affine model. Thus, we parameterize our generative quantile network as an affine latent model,

$$\varphi_X(U) = \varphi(U) + b(U)^\top f(X) \tag{12}$$

where $\varphi(U) : \mathbb{R}^d \to \mathbb{R}$ and $b(U) : \mathbb{R}^d \to \mathbb{R}^n$ are smooth functions and $f : \mathbb{R}^m \to \mathbb{R}^n$. We parameterize $\varphi$, and $b$ as neural networks with smooth activation functions while $f$ is any task-specific mapping to yield latent vector $f(X)$. By learning a non-linear transformation of $X$, high-order moments between components of $X$ can be modelled, thereby leading to a more expressive conditional generative quantile model.

Practically, dealing with uniform samples from the unit hyper-cube is less convenient computationally, due to its bounded domain. Empirically we also found the model to produce lackluster generated samples. Instead, we pre-process the uniform samples using a bijection $\Phi^{-1} : [0, 1]^d \to \mathbb{R}^d$, notably the inverse CDF of a standard normal applied component-wise. Now, the support of the model is unrestricted, which is easier to handle computationally.

In the conditional setting parameterized via the affine latent model eq. (12), problem (8) from the previous section now becomes

$$\inf_{(\varphi,b)} \int_{\mathbb{R}^d} \varphi \mathrm{d}\mu_d(U) + \int_{\mathbb{R}^d} \sup_{U \in \mathbb{R}^d} \{U^\top Y - \varphi(U) - b(U)^\top f(X)\} \mathrm{d}\nu_d(Y), \tag{13}$$

subject to zero-mean decorrelation constraints in eq. (9). The constraint can be enforced by applying batch normalization to the $f(X)$ vector without the translation term such that the empirical batch mean of $f(X)$ is zero-centered. We also find that enforcing this constraint is necessary for numerical stability and theoretical guarantees. Moreover, the supremum is computed across the batch. As we show in the following section, the dual admits a solution such that $\varphi_X(U)$ is convex almost surely with respect to $U$.

## 4 THEORETICAL ANALYSIS

Existing work in generative modeling that optimize the Kantorovich dual problem rely on parameterizing their network as an input convex neural network. Such a parameterization greatly limits the

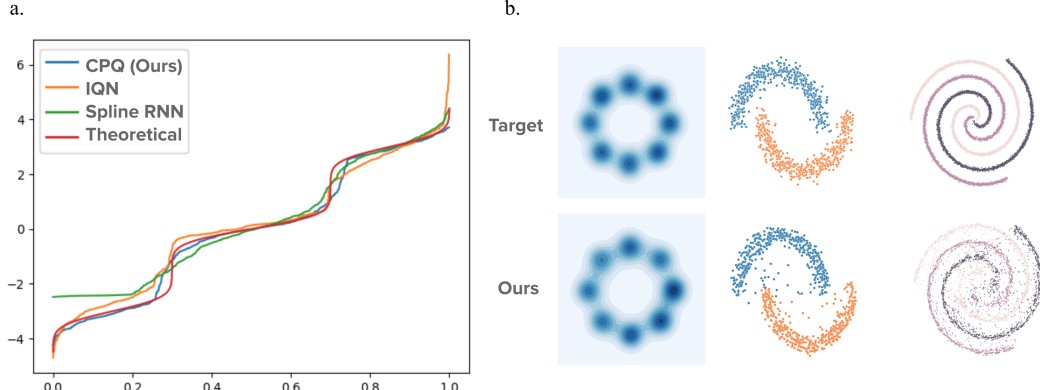

Figure 2: **Qualitative results on toy datasets.** (a.) Estimated quantile curves of the three-component Gaussian mixture model. (b.) Density estimation on `Eight Gaussians`, `Half Moons`, and `Spirals` with 1, 2, and 3 classes respectively.

architecture and expressivity of the model. We suggest that a convex parameterization of the model is not necessary when optimizing the dual problem of our correlation maximization primal. Rather, the model only needs to be smooth which can easily be achieved by using a smooth activation function. The following theorems prove that the solution to the dual problem will be convex, and the gradients of the model w.r.t. $U$ is the generative model.

**Theorem 2** (Carlier et al. 2017). *Assume $U \in \mathbb{R}^d$ is random vector with distribution $\mu$, $(X, Y) \in \mathbb{R}^m \times \mathbb{R}^d$ is random vector with joint distribution $\nu$. Furthermore, assume $\mathbb{E}(X|U) = \mathbb{E}(X) = 0$. If there exists smooth function $\varphi : \mathbb{R}^d \to \mathbb{R}$ and smooth function $b : \mathbb{R}^d \to \mathbb{R}^m$ such that $\varphi_x(u) = \varphi(u) + b(u)^\top x$ is convex for $\mathrm{Law}(X)$-almost every value of $x$ such that $Y = \nabla\varphi(U) + \nabla b(U)^\top X$, then $U$ solves the correlation maximization problem (7).*

The following theorem suggests the existence of a solution to the dual problem (13) under mild assumptions on $\nu$.

**Theorem 3** (Carlier et al. 2017). *Let $\nu$ be an absolutely continuous probability measure over $\mathbb{R}^m \times \mathbb{R}^d$ with density $g$. Assume the support of $\nu$ is $\bar{\Omega}$ where $\Omega$ is an open bounded convex subset of $\mathbb{R}^m \times \mathbb{R}^d$, and $g$ is bounded on $\Omega$ and bounded away from zero on compact subsets of $\Omega$. Then, the dual problem admits at least one solution.*

**Theorem 4** (Carlier et al. (2017)). *Let $U \in \mathbb{R}^d$ be a solution to (7) and let $\Psi : \mathbb{R}^m \times \mathbb{R}^d \to \mathbb{R}, \varphi : \mathbb{R}^d \to \mathbb{R}, b : \mathbb{R}^d \to \mathbb{R}^m$ be solutions to the corresponding dual problem (13). Let $\varphi_x(t) = \varphi(t) + b(t)^\top x \ \forall (t, x) \in [0, 1]^d \times \mathrm{support}(\mathrm{Law}(X))$. Then, $\varphi_X(U) = \varphi_X^{**}(U)$ and $U \in \partial\varphi_X^*(Y)$ almost surely.*

By the Fenchel-Moreau theorem (Rockafellar, 1970) and Theorem 4, $\varphi_X(U)$ is convex almost surely. To summarize, the aforementioned theorems establish the existence of a solution to (13), and further indicate that the gradient of the optimal convex function in the dual objective solves the primal correlation maximization problem. Lastly, Brenier's theorem establishes the foundation for parameterizing our generative model as a convex potential gradient.

Hence, the transported mass from $u \in \mathbb{R}^d$ given $x \in \mathbb{R}^m$ to $y \in \mathbb{R}^d$ is given by $y = \nabla\varphi_x(u) = \nabla\varphi(u) + \nabla b(u)^\top x$. Due to convexity of $\varphi_x$ w.r.t. $u$, $\nabla\varphi_x$ is comonotonic thereby automatically satisfies the non-crossing property of quantile networks.

**Proposition 1.** *If $\varphi_x(U) : \mathbb{R}^d \to \mathbb{R}$ is convex w.r.t. $U \in \mathbb{R}^d$ and $U_k \leq V_k$, then $[\nabla\varphi_x(U)]_k \leq [\nabla\varphi_x(V)]_k$ for all $k = 1, \ldots, d$.*

Therefore, convex potential gradients serve as a natural parameterization of quantile networks as additional constraints to enforce the non-crossing property of quantile functions are not needed. Furthermore, we prove that any conditional quantile function $Q_x(u)$ can be reasonably approximated by the affine parameterization in eq. (12).

Table 1: Performance of our work against previous one-dimensional quantile networks in fitting the three-component Gaussian Mixture Model. Results are averaged over 5 runs of 10 epochs each with same hyperparameters. Lower score is better. We use boldface for the lowest score.

| Model | MeanAE | MaxAE | sMAPE | RMSE | QL |
|---|---|---|---|---|---|
| Spline Quantile | 0.395 | 1.540 | 0.378 | 0.523 | 0.570 |
| IQN | 0.350 | 1.145 | 0.463 | 0.428 | 0.430 |
| CPQ (Ours) | **0.283** | **0.775** | **0.309** | **0.332** | **0.352** |

**Proposition 2.** *For any continuous conditional quantile function $Q_x(u)$ we can find large $n$ and functions $f(x) : \mathbb{R}^m \to \mathbb{R}^n$, $\varphi(u) : \mathbb{R}^d \to \mathbb{R}$ and $b(u) : \mathbb{R}^d \to \mathbb{R}^n$ such that the gradient of $\varphi_x(u) := \varphi(u) + b(u)^\top f(x)$ approximates $Q_x(u)$ uniformly over any compact region of $(u, x)$.*

In our experiments we find that setting $n = m$ suffices to obtain reasonable results. It would be interesting to explore in future work when and how the approximation can be substantially tightened for certain classes of conditional quantiles and network architectures for parameterizing $b$ and $f$.

## 5 EXPERIMENTS

In this section, we evaluate our proposed approach to conditional generative modeling on a variety of experiments. We first evaluate the efficacy of our proposed conditional generative quantile network as a general conditional generative model on synthetic data, MNIST (LeCun et al., 1998), and CelebA (Liu et al., 2015). Then, we evaluate the model as a conditional quantile network for probabilistic time-series forecasting on `UCI Appliances Energy` and `Google Stocks` data. The architecture we use for all experiments for our generative quantile network is smooth feed-forward neural networks to parameterize both $\varphi$ and $b$. The activation of choice must be smooth, so we choose to use CELU activation,

$$\text{CELU}(x) = \max(0, x) + \min(0, \alpha(\exp\left(\tfrac{x}{\alpha}\right) - 1)). \tag{14}$$

for some $\alpha > 0$. Unless explicitly stated, $\varphi$ has 3 layers while $b$ has 2 layers. Furthermore, we define $f(X) = X$ unless explicitly stated. For evaluation, we compute the max and mean absolute error (MaxAE and MeanAE), the $50^{th}$ and $90^{th}$ quantile loss (QL50 and QL90), the root mean squared error (RMSE), and the symmetric mean absolute percentage error (sMAPE). Full details on implementation details can be found in Appendix appendix B.2. The code for our method and all experiments will be made publicly available upon acceptance.

### 5.1 TOY EXAMPLES

The efficacy of our proposed method as a quantile function and conditional generative model are first examined on synthetic data. We consider fitting our model to learning a quantile curve, and then evaluate density estimation on 2D distributions with class-labels.

#### 5.1.1 FITTING GAUSSIAN MIXTURE MODEL QUANTILE FUNCTION

To evaluate our method as a quantile function estimator, we generate $10,000$ time-series samples of $24$ *i.i.d.* points following a Gaussian Mixture Model $GMM(\boldsymbol{\pi}, \boldsymbol{\mu}, \boldsymbol{\sigma})$ with $\boldsymbol{\pi} = [0.3, 0.4, 0.3]^\top$, $\boldsymbol{\mu} = [-3, 0, 3]^\top$, and $\boldsymbol{\sigma} = [0.4, 0.4, 0.4]$. We train the model for 10 epochs and provide an evaluation against IQN (Gouttes et al., 2021) and Spline Quantile RNN (Gasthaus et al., 2019). Figure 2.a shows the learned quantile function of the three methods relative to the theoretical quantile curve. Performance metrics are displayed on Table 1 in fitting the quantile function of $GMM(\boldsymbol{\pi}, \boldsymbol{\mu}, \boldsymbol{\sigma})$, where our CPQ consistently outperforms the two SOTA baselines across a wide variety of metrics.

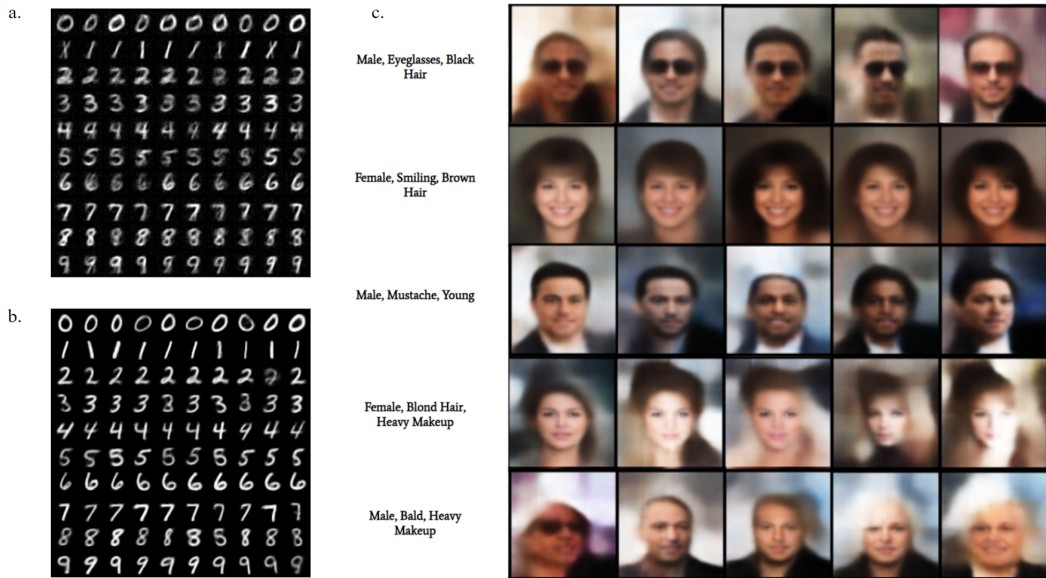

Figure 3: **Qualitative results of generated images.** For the MNIST samples, we conditionally generate the handwritten digits by conditioning on a digit for each row. (a.) Directly learning a mapping onto the image space of MNIST. (b.) Using a VAE, the quantile network maps to the latent space. (c.) Novel combination of attribute generation on CelebA.

### 5.1.2 2D DISTRIBUTIONS WITH LABELS

Here, we evaluate the efficacy of our model to generate synthetic labelled 2D distributions conditionally and unconditionally. The three densities we use are `Eight Gaussians`, `Half Moons`. and `Swirls`. The `Eight Gaussians` dataset does not distinguish points based on label, thus we use this dataset to evaluate the efficacy of our method first as an unconditional density estimator. For conditional density estimation, we turn to `Half Moons`. and `Swirls`. Each point in the `Half Moons` dataset is assigned to one of two classes, while each point in the `Swirls` dataset is assigned to one of three classes. As such, $\varphi : \mathbb{R}^2 \to \mathbb{R}^2$ for both datasets, but $b$ maps to $\mathbb{R}^2$ for `Half Moons` and $\mathbb{R}^3$ for `Swirls` with $\mathrm{spt}(b) = \mathbb{R}^2$ for both. Figure 2.b shows the generated distributions qualitatively.

### 5.2 CONDITIONAL DIGIT GENERATION ON MNIST

To further examine the ability of our method to learn distributions conditionally, we test our conditional quantile network to generate handwritten digits conditionally. The MNIST Handwritten Digits dataset contains $60,000$ gray-scale images of handwritten digits at a resolution of $28 \times 28\mathrm{px}^2$. We consider two scenarios; (1) use a variational auto-encoder (Kingma & Welling, 2014) for dimensionality reduction and generate on a latent space (Figure 3.b), and (2) directly generate on the image space (Figure 3.a). The first scenario maps $U \in \mathbb{R}^D$ to $Y \in \mathbb{R}^D$ where $D$ is the dimension of the latent space while the second scenario involves mapping $U \in \mathbb{R}^{784}$ to $Y \in [0,1]^{784}$. Clearly, the second case is a greater challenge than the first as often we set $D << 784$ and the encoder and decoder can be trained to implicitly filter out noise. However, we show that by directly generating on the image space, high quality images of the corresponding digit conditioned can be obtained, albeit slightly noisier than the auto-encoder counterpart. Nevertheless, we show that our proposed method can generate high-quality samples in strongly correlated high-dimensional spaces. Moreover, using a dimensionality reduction algorithm such as VAE can improve visual quality of the generated images by implicitly learning to filter out noise.

### 5.3 MULTI-ATTRIBUTE GENERATION ON CELEBA

Unlike MNIST, CelebA samples have multiple attribute labels rather than a single class label. We consider the model's ability to learn multi-attribute combinations and furthermore generate novel

Table 2: Performance evaluation on the multivariate time-series datasets `Energy` and `Stocks`. Results are averaged over 5 runs. Lower score is better. We use boldface for the lowest score.

| Dataset | Model | MaxAE | MeanAE | QL50 | QL90 | RMSE | sMAPE |
|---------|-------|-------|--------|------|------|------|-------|
| Energy | DeepAR | 1.033 | **0.049** | **0.024** | **0.012** | 0.103 | 0.203 |
| | TimeGAN | 0.970 | 0.129 | — | — | 0.172 | 0.378 |
| | RealNVP + RNN | 1.236 | 0.052 | — | — | 0.110 | **0.198** |
| | Huber QL | 0.980 | 0.050 | 0.025 | 0.015 | 0.094 | 0.212 |
| | CPQ (Ours) | **0.624** | 0.051 | 0.026 | 0.038 | **0.091** | 0.210 |
| Stocks | DeepAR | 0.791 | 0.067 | 0.033 | 0.018 | 0.092 | 0.408 |
| | TimeGAN | 1.379 | 0.092 | — | — | 0.181 | 0.256 |
| | RealNVP + RNN | 0.670 | 0.040 | — | — | 0.063 | **0.213** |
| | Huber QL | 0.740 | 0.026 | 0.013 | **0.007** | 0.041 | 0.220 |
| | CPQ (Ours) | **0.660** | **0.024** | **0.012** | 0.014 | **0.036** | 0.220 |

combinations of attributes. The CelebA dataset contains 202,599 RGB images of celebrity portraits aligned and cropped to $64 \times 64\text{px}^2$. Each image corresponds with a $40-$dimensional multi-hot vector that summarizes features of the celebrity portrait such as gender, hair-color, and whether the entity is wearing eyeglasses to name a few. A VAE was employed for dimensionality reduction thus our conditional quantile network maps to the latent space. Figure 3.c shows the results of novel attribute combination generation qualitatively.

### 5.4 PROBABILISTIC FORECASTING ON MULTIVARIATE TIME SERIES

Learning the entire distribution rather than predicting point-wise trajectories of trends is valuable and practical in domains like stock market analysis and demand forecasting (Wen & Torkkola, 2019). Thus, we evaluate our method on probabilistic forecasting using the `UCI Appliances Energy` dataset and `Google Stocks` data. The `UCI Appliances Energy` dataset contains 28 temporal continuous valued attributes. The `Google Stocks` dataset contains 6 temporal features gathered from 2004 to 2019. For each dataset, we generate time-series samples of 23 contiguous time points with a prediction window of 1. We compare our method against other competitive probabilistic time-series prediction methods. We use an 2-layer LSTM as the temporal feature extractor, $f$, that yields latent embeddings of the temporal covariates. For consistency, DeepAR (Gregor et al., 2014), RealNVP + RNN (Rasul et al., 2021), Huber QL, and CPQ all utilize the same backbone LSTM with same hyper-parameters. Comparisons are summarized in Table 2, where our CPQ yields comparable or superior performance and indicates more robustness across datasets and metrics than baselines.

TimeGAN (Yoon et al., 2019) and RealNVP+RNN cannot be probed to extract the $50^{th}$ and $90^{th}$ quantile levels without empirical estimation through MCMC which is computationally infeasible. Here, it is clear the advantage of a quantile approach to distribution estimation as the exact point-wise quantile level can be extracted in a single forward-pass by exploiting a comonotonic parameterization of the density estimator.

## 6 CONCLUSION

We have proposed a novel method for conditional generative quantile modelling. In particular, we construct a dual formulation of the correlation maximization problem subject to the zero mean-independence constraint along with an affine latent parameterization. Furthermore, we prove that by optimizing smooth neural networks, the optimum is convex almost surely. As such, we construct a Brenier map as the gradients of the affine latent model. Through experiments on synthetic data, MNIST, CelebA, and time series forecasting datasets, we demonstrate our method is competitive and versatile as a distribution estimator and conditioner. For future work, we would like to explore higher order parameterizations as well as other optimal transport formulations. Applications to natural language modelling and genomic data would also be of great interest.

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
