# OpenReview forum: "Conditional Generative Quantile Networks via Optimal Transport and Convex Potentials"
_ICLR.cc/2022/Conference — ICLR 2022 Submitted_

### Official Review · Reviewer_jUXa · 2021-10-30

**Correctness:** 4
**Technical Novelty And Significance:** 3
**Empirical Novelty And Significance:** 3
**Recommendation:** 5
**Confidence:** 4

**Main Review:**

Strengths:
* Well-defined model and theoretical guarantee
* Clear writing

Concerns:
* The motivation is unclear for me. There seems to be a vacancy model for conditional quantile generative modeling, but why should we consider this way?
* How do conditioned image generation related with quantiles? On the one hand, no quantitative results are provided for mnist and celeba. On the other hand, the quantiles are not used or specified for mnist and celeba dataset conditions.
* The results on mnist or celeba is less challenging. See [Ostrovski 2018] for the imagenet experiment.
* Do time series forecasting experiments assume conditions? Moreover, the CPQ does not have the best performance merics among the time series forecasting results.

**Summary Of The Paper:**

This paper proposes a conditional quantile generative model using optimal transport. Theoretical results show that the optimal is convex and a conditional dual objective is further introduced. Existing results shows the efficacy and versatility of this method.

**Summary Of The Review:**

This paper addresses the class conditioned problem which need more elaboration. Moreover, the experimental results are not strong enough to offset the motivation weakness. Thus I am tending to give a reject but still open to see more updates to make the above points more clear.

---

### Official Review · Reviewer_8X3c · 2021-10-31

**Correctness:** 2
**Technical Novelty And Significance:** 3
**Empirical Novelty And Significance:** 2
**Recommendation:** 3
**Confidence:** 2

**Main Review:**

A large part of the introduction is dedicated to a presentation of classical facts regarding quantile functions and optimal transport, including basic definitions of the pinball loss, of the Kantorovich dual problem, etc. This introduction looks a bit lengthy to me, but I must acknowledge that it is a good read with good references and great clarity.

The paper really starts its contribution with section 3: "convex potentials as generative quantile networks" and I must say that this is precisely where clarity drops in my opinion. Although I could say that I already have some experience in these topics, these parts were particularly unclear to me and definitely do not provide sufficient readability.

Considering that the majority of the paper is extremely unclear to me, although I have some experience on the topic, I suppose that --up to some nonnegligible extent at least --- it is due to an unclear exposition that combines with my non-expertise. I personally recommend rejection, but I may be willing to change my opinion if some definitely knowledgeable reviewer saw great value in the paper.


Now for my detailed review. I apologize in advance if some questions appear as trivial. Here are some questions/remarks:
* It seems that Q_\nu would be clearer than Q_Y, since Y is apparently a (vector) realization, while \nu is its law.
* in equation (5), over what exactly is expectation taken ? over Y~\nu AND U~\mu ?
* your definition 1 is quite unclear again due to the notations. Y is an element of L_d^2 or is it a vector, like a single realization of \nu  ? It looks like you are alternating between Y and \nu in some way I cannot understand. In equation (6), do we really have that \rho_\mu is a function of the single Y vector that is appearing on the right-hand side ? or is expectation taken on both U and Y~\nu with the object you want to present actually being \rho_\mu(\nu) ? This looks unclear to me.
* Maybe I am mistaken, but it looks like in practice you are using E[max(U^\top Y)] and not max(E[U^\top Y)] in your algorithm, when you compute the max per batch. Am I wrong ?
* in your theorem I, how could a vector in R^d not be square integrable ? Are you rather here talking about \nu being in L^2_d ?
* your definition 2 reads meaningless to me. Either you define Y as a realization of \nu through Y~\nu, or you define it as Y=\nabla f(U), but you can't do both. Maybe you mean that \nabla f # \mu = \nu as you write after, it just appears wrong to write it the way you do here. Furthermore, writing Y=\nabla f(U) doesn't tell me what exactly you call the convex potential quantile.
* The whole part regarding the covariate X is totally mysterious to me and I don't manage to make sense of it. First, I don't see where this covariate enters the picture in equation (7). It is a fact that U is distributed to \mu, and then what do you mean exactly with these E(X|U)=0 and E(X)=0 ? Are these further constraints imposed on X ? Then, how do you actually use X anywhere in this equation ? Then, what is it exactly you are trying to achieve with this "sup" ? This is all pretty unclear in my opinion.
* I may be missing something obvious or trivial, but I don't understand how you derive (8) ? I guess it's some special case of (4), but I don't get it. Could you please be more explicit about this, which looks like an important aspect of your paper ?
* I don't understand again how X enters the picture in (8), and I don't understand what I am seing in (9). Is it an assumption ? Some constraint you want to impose (in which case I guess that U is not distributed iid wrt \mu, since I understand that X is given)
* It reads quite strange to say that "U is pushed into standard normal" and "we pre+process the uniform samples.... component-wise". From all practical purpose, it looks to me that you are basically drawing U as gaussian i.i.d, right ? Why are you presenting things in such a complicated way ? Is there actually some reason for it ? Is theory broken if you take U~N(0,I) directly ?
* It is not clear what is your contribution when put into perspective of the theorems 2 3 and 4 you are mentioning. Why are you providing these theorems in such a detailed way if they are not a contribution ? Why are these theorems proved in the appendix ? Could you please motivate these and provide some narrative  rather than this enumeration of theorems ? I am sorry to say that this whole section is unclear to me...

* the experiments are nice, although I think it is not clear what you are exactly implementing for the time-series experiment

**Summary Of The Paper:**

That paper proposes a multivariate quantile network that may be used for generative purposes. Its contribution is to generalize some scalar characterization of quantile functions (correlation maximization) to the multivariate case and propose a dual formulation of the problem that is advocated as generalizing well.
Experiments are very limited, although somewhat convincing.

**Summary Of The Review:**

the paper apparently comes with some merit, although I must say that the exposition is so unclear that I couldn't really follow the narrative.

---

### Official Review · Reviewer_7Kv2 · 2021-11-03

**Correctness:** 4
**Technical Novelty And Significance:** 1
**Empirical Novelty And Significance:** 2
**Recommendation:** 3
**Confidence:** 4

**Main Review:**

Pros: The empirical results look encouraging. The generated pictures look realistic on both MNIST and CelebA. It also seems that the proposed method is able to avoid mode collapse and can explore different modes.

Cons:
-	The main concern of this paper is its methodological and theoretical novelty over the existing papers. A thorough comparison between the current paper and the existing works is missing. The problem formulation and theoretical results follow somewhat straightforwardly by existing papers, i.e. the cited Carlier et al. (2017). In particular, Eq. (13) is essentially Eq. (4) in Carlier et al. (2017) (with errors in the integration domain). The theorems all resemble those in Carlier et al. (2017).
o	It looks that the only novelty is in using neural networks to parameterize the potential function. What is the advantage of this method over that in Carlier et al. (2017), in which an estimation algorithm was also proposed?
o	Currently the proofs of the theorems in the supplementary materials look like a direct copy of those in Carlier et al. (2017). It may be worth pointing out if there is any technical difficulty from extending the theorems of Carlier et al. (2017) to the present paper.
o	From the pseudo code in Algorithm 1, I couldn’t find which lines are related to neural networks, although Table 3 does show the results obtained by using LSTM. There seems to be a gap between the claim in the main text and the actual algorithm, and clarification may be needed.
-	The convex potential is learned by minimizing the dual problem. In order to facilitate the use of optimizers like SGD, how to compute the gradient? It is not clear if a typical auto differentiation can work given that the dual problem involves integrals and constraints. As noted in Sect. 4.1 of Carlier et al. (2017), the dual problem can be further formulated as an unconstrained problem, which may be more readily solvable by using SGD type optimizers, and more scalable to a higher dimension. More details may be needed.
-	Given that there are many popular conditional GAN models, e.g. BicycleGAN etc. that can overcome the mode collapse problem, a more comprehensive empirical experiment is necessary to fully showcase the advantage of the proposed method.

**Summary Of The Paper:**

This paper estimates conditional quantile contours of a multivariate random vector. As shown by previous papers, his problem can be reformulated as a correlation maximization problem, and its solution exists and can be represented as the differential of a potential function by Brenier’s theorem. Solution can be found by considering the dual problem, as also shown by previous papers. This paper claims to parameterize the potential function by neural networks, which leads to a new method.

**Summary Of The Review:**

Although the empirical results are encouraging, it is unclear the method and theory in this paper is sufficiently novel. Therefore, I am as of now not in favor of recommending to accept this paper.

---

### Official Review · Reviewer_Vf6H · 2021-11-03

**Correctness:** 3
**Technical Novelty And Significance:** 1
**Empirical Novelty And Significance:** 1
**Recommendation:** 3
**Confidence:** 4

**Main Review:**

#### Novely:
Going through the two works of Carlier et al. 2016 and Carlier et al. 2017, I noticed that the paper's exposition (Sections 3.1 and 3.2) is quite similar to the last work of  Carlier et al. 2017. It would be better if the authors highlight their contributions compared to these last works.

#### Weakness:
- In Eq. (12) the authors parametrize their generative quantile network as an affine latent model using a smoothing function $f$. However, the theoretical analysis (Theorems 2, 3, 4) assumes that $f=Id$. I am wondering if the theoretical guarantees are still valid for general $f$.
- In Eq. (8), the minimization problem is over $(\varphi, \psi, b)$, however in the objective and the constraints this parameter doesn't appear. According to Eq. (3) in Carlier et al. 2017, I think the constraint must be $\phi(U) + \psi(Y) + b(U)^\top X \geq U^\top Y.$
- In Definition 2 and the beginning of Section 3.2, the same notation $\nu$ is used to denote the distribution of $Y$ variable and the joint distribution of $(X,Y)$.
- In Figure 1, it is not clear for me if Training and Inference are separate; in other words, is it an end-to-end DNN alternating between Training and Inference?

#### Clarity:
The paper is overall clear and easy to follow but contains many typos or grammar mistakes, I have listed some of them below. However, it doesn't impact readability.

#### Typos and inaccuracies:


##### Missing:
- page 4: then there exists a closed convex semi-continuous function f. The word "semi-continuous" is missing.

##### Notation:
- At the beginning of Section 2, it would be better if you use letter "Y" to define the CDF, since the pinball loss is defined using "Y", and giving the definition of notation "(.)^+" and "(.)^-"
- I suggest to write "Eq." when referring to equations instead of "eq."

##### Typos:
- page 1: "in its the univariate" -> "in its univariate"
- page 2: "esimation" -> "estimation"
- page 2: "conduct thorough" -> "conduct through"
- page 2: "bet it univariate" -> "be univariate"
- page 2: "for the quantifule" -> "for the quantile"
- page 2: "on top of an reccurent" -> "on top o a reccurent"
- page 2: "level u with a univariate" -> "level u with an univariate"
- page 3: "should the dual admit" -> "the dual should admit"
- page 3: "neither are they genrative" -> "neither are the generative"
- page 4: "Should the distribution mu" -> "the distribution mu should be"
- page 8: "Swirls" -> "Spirals"
- page 8: "turn to Half Moons." -> "turn to Half Moons"

**Summary Of The Paper:**

Quantile regression is frequently used as an alternative to conventional regression. An important advantage of quantile regression is that provides enough flexibility to capture the whole conditional distribution, rather than the conditional mean, of the response variable for given predictor variables. Standard approaches to tackle the estimation problem of quantile regression are based on the so-called "pinball loss" (quantile loss) in the univariate case, or an $L_p$-norm of the vector quantile loss in the multivariate case. This paper builds conditional generative quantile modelling to the multivariate setting. The authors propose convex potential quantile (CPQ), which is an optimal transport approach through a Kantorovih dual objective functional, which maximizes the correlation between the target distribution (dependent variable in the regression case) and the source distribution (uniform distribution over the unit hypercube). Various experiments are conducted on synthetic and real datasets, including generative modelling (MNIST and CelebA) and probabilistic forecasting.

**Summary Of The Review:**

The authors propose convex potential quantile (CPQ), which is an optimal transport approach through a Kantorovih dual objective, maximizing the correlation functional between a uniform distribution over the unit hypercube and target distribution. Their approach is heavily based on a recent work of Carlier et al. 2017. In my opinion, the experiment part of the paper highlights the beauty of the conditional vector quantile proposed in the work of Carlier et al. 2016, through neural networks implementation.

---

### Decision · Program_Chairs · 2022-01-20

**Decision:**

Reject

**Comment:**

This paper proposes a conditional quantile generative model using optimal transport. Although the problem addressed in this paper is interesting and important, the proposed convex potential quantile (CPQ) approach is highly relevant to a recent work (Carlier et al. 2017). Due to the lack of clear explanations of the contributions compared to the existing work, none of the reviewers suggested acceptance of this paper.